# Genetic Markers Associated with Milk Production and Thermotolerance in Holstein Dairy Cows Managed in a Heat-Stressed Environment

**DOI:** 10.3390/biology12050679

**Published:** 2023-05-04

**Authors:** Ricardo Zamorano-Algandar, Juan F. Medrano, Milton G. Thomas, R. Mark Enns, Scott E. Speidel, Miguel A. Sánchez-Castro, Guillermo Luna-Nevárez, José C. Leyva-Corona, Pablo Luna-Nevárez

**Affiliations:** 1Departamento de Agricultura y Ganadería, Universidad de Sonora, Hermosillo 83000, Mexico; 2Department of Animal Science, University of California, Davis, CA 95616, USA; 3Texas A&M AgriLife Research, Beeville, TX 78102, USA; 4Department of Animal Sciences, Colorado State University, Fort Collins, CO 80523, USA; 5Departamento de Ciencias Agronómicas y Veterinarias, Instituto Tecnológico de Sonora, Ciudad Obregón 85000, Mexico

**Keywords:** GWAS, heat stress, Holstein cows, milk, SNPs, thermotolerance

## Abstract

**Simple Summary:**

Holstein is the most popular dairy cattle breed worldwide due to its milk yield. When these cows are exposed to heat stress, they reduce feed intake and milk production in order to minimize body heat production. The large variability associated with this response appears to be genetically regulated. Therefore, we combined genomic and marker-assisted technologies with the objective to validate genetic markers associated with milk production and thermotolerance. A genome-wide association study detected six candidate single nucleotide polymorphisms (SNPs) as predictors for milk production in heat-stressed Holstein cows. Only three of these SNPs were further validated as markers for milk production and thermotolerance traits (i.e., rectal temperature and respiratory rate) in two independent Holstein cow populations. Such markers belong to genes that regulate metabolic functions needed to accomplish energy demands and minimal heat production. The results of this study revealed that heat-stressed Holstein cows with favorable markers were able to reduce rectal temperature and respiratory rate, which allowed them to maintain adequate milk production levels. In conclusion, we validated three genetic markers in heat-stressed Holstein dairy cows, which are useful to be included in selection programs to improve milk yield and tolerance to heat stress.

**Abstract:**

Dairy production in Holstein cows in a semiarid environment is challenging due to heat stress. Under such conditions, genetic selection for heat tolerance appears to be a useful strategy. The objective was to validate molecular markers associated with milk production and thermotolerance traits in Holstein cows managed in a hot and humid environment. Lactating cows (*n* = 300) exposed to a heat stress environment were genotyped using a medium-density array including 53,218 SNPs. A genome-wide association study (GWAS) detected six SNPs associated with total milk yield (MY305) that surpassed multiple testing (*p* < 1.14 × 10^−6^). These SNPs were further validated in 216 Holstein cows from two independent populations that were genotyped using the TaqMan bi-allelic discrimination method and qPCR. In these cows, only the SNPs rs8193046, rs43410971, and rs382039214, within the genes *TLR4*, *GRM8*, and *SMAD3*, respectively, were associated (*p* < 0.05) with MY305, rectal temperature (RT), and respiratory rate. Interestingly, these variables improved as the number of favorable genotypes of the SNPs increased from 0 to 3. In addition, a regression analysis detected RT as a significant predictor (R^2^ = 0.362) for MY305 in cows with >1 favorable genotype, suggesting this close relationship was influenced by genetic markers. In conclusion, SNPs in the genes *TLR4*, *GRM8*, and *SMAD3* appear to be involved in the molecular mechanism that regulates milk production in cows under heat-stressed conditions. These SNPs are proposed as thermotolerance genetic markers for a selection program to improve the milk performance of lactating Holstein cows managed in a semiarid environment.

## 1. Introduction

Milk production is a very important livestock activity because it generates a basic food for humans. Cattle account for 81% of the total milk produced worldwide [1]. The dairy industries located in semiarid regions face the challenge of producing quality milk in a warm climate, which is characterized by a high ambient temperature and humidity leading to heat stress. Installation of cooling systems to dissipate heat during summer seasons, automated feed machinery, and herd improvement using genomic-based technologies are strategies that dairy producers are preparing for the future [2].

Heat stress is an imbalance between the ratio of heat acquired from different sources such as body metabolism and environmental conditions versus the heat dissipation system, which causes an increase in the animal’s body temperature [3]. The Holstein breed of dairy cattle is the most popular worldwide due to its outstanding milk yield and ease of management. During summer, dairy cows’ ability to dissipate heat through skin evaporation is limited by their low live weight to body surface area ratio, the presence of underdeveloped sweat glands, and their short, dense body surface hair, all of which affect their milk production [4].

Fermentation of forage consumed in the rumen produces considerable metabolic heat, which also generates additional heat load for the cow’s body [5]. Additionally, enhanced warming of the climate with the consequences of the intensification of hot periods has caused dairy cows to be subjected to long periods of intense heat stress [6], which threatens to severely reduce productive performance, especially milk production [7]. Therefore, the selection of livestock adapted to semiarid regions with a superior ability to tolerate drought and extreme temperatures will be essential to ensure profitable production and cope with the challenging climate scenario expected in the coming years [8,9].

With the progress of bovine genome sequencing and high-density SNP genotyping technologies, genome-wide association studies (GWASs) have become an important tool for locating QTL or chromosomal regions associated with traits of economic importance [10]. In various dairy breeds of cattle, GWASs have been successfully employed to identify regions in DNA involved in heat tolerance [11], lipid movement [12], milk production and fertility [13], production persistency [14], and lactation curve characteristics [15]. These traits and QTLs are summarized in the Animal QTL database: milk production and yield (9121), milk composition—fat (45,625), milk composition—protein (25,916), reproduction (45,715), production traits (22,722), meat and carcass (22,695), exterior (10,388), and health (including heat tolerance; 8242) (QTLdb; http://www.animalgenome.org/QTLdb. Date accessed: 23 April 2023).

GWASs examine thousand of small variants across the genome called single-nucleotide polymorphisms (SNPs), which are very helpful to identify functional genes likely involved in phenotypic traits [16]. For instance, the SNP K2323A from the gene DGAT1 has been associated with genetic variation in milk yield and composition, as this SNP decreased milk production and increased milk fat, then showed a genetic marker effect of the SNP K232A on milk traits [17,18].

Recently, GWASs have been performed in Holstein cows subjected to heat stress conditions. Potential candidate genes involving the heat stress response have been discovered as associated with milk production traits [19]; physiological traits such as rectal temperature, respiratory rate, drooling score [20,21]; and milk fatty acid composition [22]. The identification of specific genes or SNP markers inferring thermotolerance in Holstein cows could be an important strategy to ensure dairy cows' productivity [23]. However, these markers must be validated in independent cattle populations to be available for incorporation into genetic selection programs focused on improving complex and economically important traits such as milk production and thermotolerance.

Therefore, our objective was to validate candidate genomic SNPs as molecular markers associated with milk production and physiological traits indicative of thermotolerance in Holstein cows managed in a hot and humid environment.

## 2. Materials and Methods

The Institutional Animal Care and Use Committee of the Instituto Tecnologico de Sonora approved all procedures performed on animals (approval code 2017-02).

### 2.1. Experimental Locations and Animals

The study was conducted at three neighboring dairy farms located in the Yaqui Valley, Sonora, Mexico (27°21′ N 109°54′ W). These farms were managed under an intensive system for milk production. Climatic conditions throughout the year were characterized by significant variations in temperature (−2.5 to 45.9 °C), relative humidity (25 to 90%), and solar radiation (0.05 to 761.7 W/m^2^). Annual precipitation was ~341 mm/year ranging from 3 to 90 mm on rainy days.

Three hundred spring-calved Holstein cows, about 4 to 6 years old, average body weight of 645.1 ± 32.5 kg, and average body condition of 3.8 ± 0.01, were used in this study. Selection criteria included age, body weight, body condition, and calving season because spring-calving cows experience heat stress during their peak milk production period.

Similar management was provided to all cows, which were housed in shaded barns with free access to a water source and a commercial supplement of trace minerals. The floor and shade spaces were provided as per the need of Holstein cows. In order to supply the nutritional requirements for dairy cattle in lactation, cows received a mixed ration twice a day that was formulated considering an average weight of 650 kg and milk production of ~30 kg/day, with an average composition of 3.5% fat and 3.2% true protein [24].

### 2.2. Milk Yield Records and Climatic Data

Cows were milked twice per day (0700 and 1700) and milk production (MY; kg) was recorded daily using an electronic system (Metraton 21TM, Westfalia-Surge Farm Technologies, Siemensstraße, Bönen, Germany). A total of 183,000 milk records were collected from 300 complete lactations. Data were filtered by the exclusion of wrong records (e.g., 5 L, 80 L) and then used to calculate the adjusted 305-day milk yield (MY305) per cow in kg. The MY305 was calculated by multiplying milk production levels by an adjustment factor provided by the Dairy Herd Improvement Association, which included days in milk and the age of the cow [25]. These Holstein cows were daughters of 93 purebred bulls and 265 dams.

Ambient temperature (AT; °C) and relative humidity (RH; %) data were collected by a nearby meteorological station (Network of Automatic Meteorological Stations of Sonora) and extracted from the REMAS website (http://www.siafeson.com/remas; accessed on 8 April 2022). Daily records of AT and RH were used to calculate the temperature–humidity index (THI) using the formula THI = (0.8 × AT) + [(RH/100) × (AT − 14.4)] + 46.4 [26].

### 2.3. Genotyping and Quality Control

Disposable sterile syringes were used to collect from each cow a blood sample (3 mL) through venipuncture of the coccygeal vein. Five drops of whole blood were spotted on Fast Technology for Analysis of Nucleic Acids cards (FTA^®^), which were sent to Neogen AgriGenomics (Lincoln, NE, USA) for DNA extraction. DNA was genotyped using the SNP panel BovineSNP50 containing 53,218 highly informative SNPs uniformly distributed across the entire bovine genome (Illumina Inc., San Diego, CA, USA) [27].

Quality control was performed using PLINK software to remove all SNPs that did not comply with the following criteria: (1) minor allele frequency of an SNP greater than 10% (MAF > 0.10), (2) no deviation from Hardy–Weinberg equilibrium (*p*-value of chi-square goodness-of-fit test greater than 0.05, X^2^ > 0.05), (3) call rate of an individual genotype higher than 95%, and (4) call rate of a single SNP genotype higher than 90%. Additionally, the SNPs located on the sex chromosome or those with unknown positions were removed from the current study. After quality control measures, 43,691 SNP markers were retained for further analyses.

### 2.4. Genome-Wide Association Analysis (GWAS)

The principal component analysis (PCA) option was used to correct batch effects/stratification of the test input data. The GWAS was performed using a single-locus mixed model to study associations between genotypes of each SNP marker with the trait MY305 (i.e., single-marker SNP GWAS), by fitting all SNPs simultaneously. The additive, single-locus, mixed model used was: y = Xβ + Za + e, where **y** was the vector of phenotypic observations (MY305), X was the design matrix of fixed effects, Z was the design matrix of random additive genetic effects, β was the vector of fixed effects, a was the vector of random additive genetic effects, and e was the vector of residual effects. It was assumed that a~*N* (0, Gσ^2^_a_) and e~N (0, Iσ^2^_e_), where σ^2^_a_ is the additive genetic variance, σ^2^_e_ is the residual variance component, G is the genomic relationship matrix, and I is the identity matrix.

The software SNP & Variation Suite v8 (SVSv8; Golden Helix, Inc., Bozeman, MT, USA, www.goldenhelix.com; accessed on 12 June 2022) was used to perform single-marker GWAS analysis. The statistical model included genotype, herd, and lactation number as fixed effects, days in milk as the covariate, and sire as the random term to account for family effects. A total of 93 Holstein bulls were included as sires in the current study.

### 2.5. Multiple-Testing Correction

The Bonferroni correction test (b = α/n), which assumed independence between SNPs, was used to adjust and set for multiple comparisons the *p*-values for the SNPs obtained from GWAS analyses [28]. The experiment-wise error was α = 0.05, and the number of tests (n) was taken to be the number of the useful SNPs (*n* = 43,691). Therefore, the corrected *p*-value threshold for the genome-wide significant level was 1.18 × 10^−6^ (i.e., 0.05/43,691), which corresponded to 5.93 on a −log10(*p*-value) scale.

### 2.6. Candidate Genes and Pathway Analyses

The functional genes that contained or were located less than 0.1 MB of distance from the significant SNPs associated with MY305 were selected as candidate genes. The genome assembly *Bos*_*taurus*_UMD_3.1.1. was used as a reference genome to search for candidate genes according to the chromosomal locations of each significant SNP.

Functional pathway enrichment analysis was performed using candidate genes that were significant at a *P* less than 0.01 (*n* = 36) and the bioinformatics online database KOBAS (http://kobas.cbi.pku.edu.cn/; accessed on 12 August 2022). Ensembl (https://useast.ensembl.org/index.html; accessed on 23 July 2022) and GeneCards (http://www.genecards.org; accessed on 15 August 2022) were used for gene descriptions (i.e., registration code, name, and chromosomal location).

### 2.7. SNP Validation Study

Two independent cattle populations including 216 Holstein cows (*n* = 112, *n* = 104) from neighboring dairy farms were used to validate, as molecular markers for milk yield and thermotolerance, three SNPs previously detected as associated (*p* < 0.05) with milk yield in Holstein cows exposed to heat stress. These SNPs were rs8193046, rs43410971, and rs382039214, which were within the genes *TLR4* (toll-like receptor 4), *GRM8* (glutamate metabotropic receptor 8), and *SMAD3* (SMAD family member 3), respectively.

Cows included in the validation study were spring-calved, 3 to 6 years old, and body conditions ranged from 2.5 to 4.0. Milk records were collected daily from each cow through the entire lactation using an electronic system and were adjusted to obtain the individual MY305. Physiological traits of rectal temperature (RT) and respiratory rate (RR) were measured biweekly (06:00 and 16:00 h) as indicators of thermotolerance. A digital thermometer (TES-1310R) with a contact sounding line (Type K; -com large) that touched the rectal mucus was used to record RT, whereas RR was evaluated by visual counting of the flank intercostal movements for 60 s (breaths/min). Both RT and RR were evaluated in the waiting parlor before milking, once the cow remained quiet to facilitate collection of physiological data by the same person all the time.

Vacutainer tubes that contained the anticoagulant disodium ethylenediamine tetraacetic acid (EDTA-Na2; Venoject^®^, Terumo, Lakewood, CA, USA) were used to collect a blood sample from each cow by puncturing the jugular vein. After placing in a thermal cooler, the samples were transported to the Instituto Tecnologico de Sonora and centrifuged at 3500 RMP for 15 min. A pipet was used to collect 200 μL of leukoplatelet layer from each sample, which was refrigerated at −20 °C before proceeding with DNA extraction. A commercial kit was used for DNA extraction (DNeasy Blood & Tissue Kits; Cat. 69504, QIAGEN, Hilden, Germany) according to the manufacturer’s instructions. A NanoDrop automatic spectrophotometer (NanoDrop, Thermo Fisher Scientific, Waltham, MA, USA) was used to measure the concentration (260 nm absorbance) and purity (260/280 ratio) of DNA obtained. Then, electrophoresis in 1% agarose gel stained with 1.5 μL ethidium bromide was used to confirm the integrity of the DNA.

The TaqMan method for allelic discrimination and RT-qPCR (StepOneTM, Applied Biosystems, Foster City, CA, USA) were used to genotype the four significant SNPs reported in our study, according to the procedures described by Castillo-Salas et al. [29]. The StepOne Real-Time PCR System (Thermo Fisher Scientific, Waltham, MA, USA) was used to perform the PCR. Finally, the StepOne software (version 2.3, Life Technologies Corporation, Carlsbad, CA, USA) was used for PCR data analysis and genotyping.

### 2.8. Statistical Analyses

The procedure MEANS was used to calculate descriptive statistical measures for the traits MY305, RT, and RR. Similarly, the procedures UNIVARIATE and GLM (Levene’s test) were performed to evaluate the assumption of normality of data distribution and equality of variances, respectively. Then, PROC ALLELE was used to calculate both allele and genotype frequencies. A chi-square (X^2^) test was performed in PROC FREQ to analyze Hardy–Weinberg equilibrium (HWE). The SAS software (Version 9.4; SAS Inst. Inc., Cary, NC, USA) was used to perform all statistical procedures.

An associative study between genotype and phenotype was conducted to validate three genomic SNPs as molecular markers for milk production and thermotolerance in two independent Holstein cattle populations under heat stress. This study was conducted after verifying that the data had a normal distribution within each validation dairy population. Statistical mixed procedures were used to test the candidate SNPs as predictors for the traits MY305, RT, and RR. The statistical model included the response variable; the SNP genotype, herd, and lactation number as fixed effects; days in milk as the covariate; and sire as the random effect.

Preplanned pairwise comparisons of least-squares means were generated using the PDIFF option when the associative analyses detected the genotype term as a significant source of variation (*p* < 0.05). The option LSMEANS with Bonferroni adjustment was used to process these means separation tests [30]. The MIXED procedure in SAS was used to calculate the estimated effects of the average allele substitution, which were generated by regressing the phenotype on the number of copies of one SNP’s allele as a covariate (e.g., the effect of substituting 1 allele with another allele) [31]. Additive and dominance (e.g., non-additive) genetic effects were calculated in SAS following the procedures described by Falconer and Mackay [32].

The data from the validation population were represented in tables with the average values ± standard errors according to SNP genotypes. Data were also represented in figures that included average values ± standard errors according to the number of SNPs with favorable genotypes.

### 2.9. Gene Marker Effects on Milk Production and Thermotolerance Traits

One-way ANOVA was used to compare MY305, RT, and RR in cows according to the number of favorable SNP genotypes. Data processed for this analysis belonged to a normal and homoscedastic population. The Tukey HSD test was performed for pairwise comparisons. For these analyses, statistical significance was declared at *p* < 0.05. In order to confirm the thermotolerance effect of the SNP markers identified in this study, a correlation between MY305 and physiological traits was performed using PROC CORR within cow groups with different numbers of favorable SNP genotypes. Finally, a linear regression analysis including RT or RR as predictors for MY305 was performed using PROC REG in cow’s groups with 0, 1, and >1 favorable SNP genotypes.

## 3. Results

### 3.1. Climatic Conditions

According to ambient temperature and relative humidity data, the THI through the study averaged 68.5 units in late spring, ranged from 70 to 82 in the summer, and averaged 72.6 units in fall. These THI values suggested environmental conditions leading to heat stress in dairy cows involved in the current study.

### 3.2. Whole-Genome Association Study

The single-marker GWAS identified 18 SNPs associated with the trait MY305 at a significant *p*-value less than 0.001. Only six of these SNPs surpassed Bonferroni multiple-testing corrections because they showed a *p*-value less than the threshold 1.14 × 10^−6^, as presented in Figure 1. The SNPs rs8193046, rs43410971, and rs382039214 were intronic variants located within the genes *TLR4*, *GRM8*, and *SMAD3*, respectively. The SNP rs109479519 was near to *GLRX5* gene (0.032 Mb), whereas the SNPs rs29015299 and rs108988401 were intergenic but more than 0.1 MB from the nearest gene (Table 1).

### 3.3. Functional Enrichment Analyses

Functional pathways for the candidate genes associated with MY305 at *p* < 0.01 are presented in Table 2. Only pathways that were significant (*p* < 0.05) after Benjamini–Hochberg correction were retained for the validation study. Genes involved in such pathways were *TLR4*, *SMAD3*, *GRM8*, *CARD11*, *TCL1A*, *NFKB1, SMAD6*, and *TRPC1*.

### 3.4. SNP Marker Association Study

Of the six SNPs that were associated with MY305, only four of them were retained for further study because they were within or near a candidate gene. However, only three out of these four SNPs were accomplished with the criteria for minor allele frequency higher than 10% (MAF > 0.10) and no deviation from the Hardy–Weinberg equilibrium (HWE, X^2^ > 0.05). These 3 SNPs were rs8193046, rs43410971, and rs382039214 within the genes *TLR4*, *GRM8,* and *SMAD3*, respectively (Table 3), and they were designated as suitable to be analyzed in additional genotype to phenotype association and validation studies.

The least-square means for milk yield at 305 d and physiological traits are reported in Table 4. The SNPs rs8193046 and rs43410971 were associated with MY305, RT, and RR (*p* < 0.001), whereas the SNP rs382039214 was associated with MY305 and RT (*p* < 0.001).

The most favorable genotypes for the SNPs rs8193046, rs43410971, and rs382039214 were AA, GG, and TT, respectively, because they were associated with higher milk production and more beneficial measurements of RT and RR in cows. Combined with genotype average values, these results appeared to confirm the favorable effect of the genes *TLR4*, *GRM8,* and *SMAD3* on milk production and physiological traits indicative of thermotolerance in Holstein cows exposed to a heat-stressed environment.

### 3.5. Allele and SNP Genotype Effects on Phenotypic Traits

Effects of allele substitution and fixed estimates are presented in Table 5. The SNPs rs8193046 and rs43410971 had the highest allele contribution (*p* < 0.01) for MY305 and RR, whereas the SNP rs382039214 was the most beneficial contributor for RT. In addition, an additive fixed effect was confirmed (*p* < 0.01) for these SNP markers, suggesting that the sum of their individual effects was equal to their combined allele effects.

Further genotype analysis confirmed a significant improvement (*p* < 0.05) in MY305 (Figure 2a) as the number of favorable genotypes of the three significant SNP markers increased from 0 to 3 in Holstein cows. Similarly, RT and RR also improved (*p* < 0.05) as the number of favorable genotypes increased (Figure 2b,c).

### 3.6. SNP Markers Effects on Relationship between Milk Yield and Thermotolerance

Pearson correlation analysis showed a moderate association between MY305 and RT in cows carrying 0 and 1 favorable genotype (r = −0.4587 and r = −0.5345, respectively; *p* < 0.01). Interestingly, this relationship increased in cows carrying more than one favorable genotype (r = −0.6021; *p* < 0.01). However, a low relationship was detected between MY305 and RR in cows from all favorable genotype groups (*p* < 0.05; Table 6).

Linear regression analysis identified RT as a significant predictor for MY305 (*p* < 0.01). The regression coefficient (β_1_) improved as the number of SNPs with favorable genotypes increased, as well as the coefficient of determination (R^2^). These results suggested a change of −140.35, −277.51, and −470.39 L in MY305 per unit of change in RT in cows carrying 0, 1, and >1 favorable SNP genotypes, respectively (*p* < 0.01; Figure 3a–c).

## 4. Discussion

An environmental factor that exerts a substantial negative effect on dairy performance is heat stress, which results when the internal heat production is higher than the cow’s body’s ability to dissipate it [4]. In recent times, lactating dairy cows have been exposed to even more severe heat stress due to the steadily warming global climate, the lengthening of drought periods, and the intensification of the warm season throughout the year [6]. Moreover, the intensive selection for high milk-yielding Holstein cows that are subjected to a tremendous metabolic heat load has impaired these cow’s ability to maintain homeothermy, especially when they are exposed to severe heat stress conditions, making these cows more sensitive or susceptible to environmental changes [33,34].

The Yaqui Valley in Sonora Mexico is characterized by extended periods of high ambient temperature and high relative humidity throughout the year, which elevates the THI value above the threshold of 68 units in the middle spring. As the warm environmental conditions become more intense, the THI begins to increase steadily in late spring, peaking in the summer and then declining in fall. This very hot weather is unique in northwest Mexico and creates adverse heat stress conditions for cattle. This location provides an opportunity to study the genetic basis of thermotolerance in lactating Holstein cattle. However, only a few dairy farmers are willing to collaborate in this type of research because of the decrease in milk production of the cows under study.

Genomic technologies and marker-assisted selection programs have been used as successful strategies to study candidate genes and genetic markers associated with heat stress response in Holstein cows [35]. In the current study, we performed both technologies sequentially in order to identify and validate SNPs as potential molecular markers to be used in genetic selection and herd improvement programs. After executing GWAS, we identified 18 SNPs as predictors of milk production performance in heat-stressed Holstein cows. Four of these SNPs were found to meet with the selection criteria and were genotyped in independent Holstein populations using the TaqMan molecular assay to obtain the SNP genotypes. From these SNPs, only three were tested through a genotype-to-phenotype association study and validated as molecular markers for milk production and thermotolerance traits. Further analyses confirmed the favorable effect of these markers on milk production and physiological traits indicative of heat stress tolerance (i.e., rectal temperature and respiratory frequency), as well as the positive influence of the markers on the predictive relationship between rectal temperature and milk production.

A genetic component has been reported to influence milk production in dairy cows exposed to heat stress, suggesting the presence of thermotolerance-related genes [19,36]. The response or sensitivity to heat stress in milking cows appears to be associated with pronounced chromosome-wide genetic variances, indicating that genomic regions and major candidate genes should underlie the heat stress response [37]. Identification of molecular markers or SNPs associated with tolerance to heat stress has been proposed as an important strategy to ensure productivity in cattle exposed to thermal stress [38]. Therefore, the study of the genetic basis associated with thermotolerance is potentially helpful for the selection of cattle adaptable to warm semiarid conditions [39].

In the current study, single-marker GWAS analysis detected 18 SNPs associated with milk production in heat-stressed cows. Of these SNPs, only six of them surpassed multiple-testing selection criteria and were proposed as genomic candidate SNPs for milk production in Holstein cows, which were managed in a warm semiarid environment from northwest Mexico (i.e., Yaqui Valley). This low number of significant SNPs may have been influenced by the small sample size. As GWASs involve thousands of SNP markers, a large sample size is usually required to reach appropriate statistical power. However, when a small sample is available, GWASs may be limited to detecting true associations between SNPs and phenotypes leading to increase false negative rates [16].

Genome-wide association studies (GWASs) have become a widely used approach to identify genomic regions and genetic variants associated with phenotypes of interest [10]. A GWAS performed for milk yield in heat-stressed Holstein cows identified three genomic regions harboring candidate genes involved in the cellular response to heat stress [19]. Chromosomal regions and candidate genes associated with milk fatty acid composition were reported in Holstein cattle also subjected to a GWAS for heat stress response [33]. Similar GWAS analyses in Holstein cows identified non-overlapping genomic regions and candidate genes associated with physiological heat stress indicators and confirmed the polygenic nature of heat tolerance, as well as complementary mechanisms associated with heat stress response [21,40].

A QTL with major effects on milk production traits and milk fatty acid profiles has been detected in the centromeric region of cattle chromosome 14 (BTA14) [22]. Although several genes with major effects on milk yield and milk fat have been reported, the major contribution to the genetic variance of BTA14 associated with these milk traits is attributed to the DGAT1 gene [41]. According to Grisart et al. [42], a nonconservative lysine to alanine (K232A) mutation in the DGAT1 gene explained daughter deviations for milk yield (18%), fat yield (15%), and protein yield (8%).

Research involving DGAT1 mutation K232A (Lys232 → Ala) has focused on the study of milk fat production because it encodes the enzyme diglyceride O-acyltransferase 1 (DGAT1), which catalyzes the synthesis of triglycerides in the mammary gland. However, this SNP is also involved in the structure of mammary gland epithelial cells suggesting its influence on milk synthesis [43]. The DGAT1 K232A mutation has been reported to be associated with an increase in milk fat production and a reduction in milk yield and milk protein in New Zealand [44] and German Holstein dairy cattle [17].

From the six genomic SNPs detected in our study, only three were within a candidate gene and surpassed selection criteria (i.e., MAF > 0.10 and HWE > 0.05), which were tested in a marker SNP association study. These SNPs were associated with MY305, RT, and RR, and they were proposed as molecular markers for milk production and heat stress tolerance. The results supported prior reports from GWAS about the existence of a genetic component underlying heat stress tolerance in Holstein dairy cows. However, to our knowledge, this is the first report of genomic SNPs from candidate genes associated with milk production in heat-stressed Holstein cows, which were further validated as molecular markers for milk performance and thermotolerance in independent Holstein cattle populations managed under extremely warm environmental conditions.

Genomic analyses using GWASs are able to detect markers distributed across the whole genome; it is likely to be more successful compared to the candidate-marker approach, which is generally based on testing a small number of SNP markers [45]. However, the combination of both technologies has been proposed for the identification of genomic candidate SNPs and their further validation as molecular markers in independent cattle populations [46]. This approach was also suggested to validate candidate genes identified from gene expression studies [47].

Statistical models of GWAS are able to detect individual significant associations between thousands of SNPs and a specific phenotype. In order to avoid false positives due to multiple SNP testing, a correction procedure must be applied and it results in a very high significance threshold lacking to detect polygenic effects [48]. Controlling the false discovery rate (FDR), as well as similar methods, were developed as an alternative to control the rate of the experiment-wide error in GWAS results [49,50]. However, the validation of the SNP effects in independent animal populations appears to be the most reliable procedure to test the importance of SNPs [51], because of the small probability of an SNP being significant in two different populations [52]. This information reinforces our rationale for the validation of SNPs at different dairies.

Pryce et al. [53] identified eleven genomic SNPs in three regions on chromosomes 6 and 26 associated with lactation persistency in Australian dairy cattle; these SNPs were further validated as potential genetic markers in two dairy cow populations from different breeds (i.e., Holstein and Jersey). In a similar study, Chamberlain et al. [46] identified 43 genomic SNPs on chromosome 20 associated with milk production traits, also in Australian dairy cattle, after executing three GWAS studies. These SNPs were further evaluated in a validation population that revealed molecular markers for milk protein composition.

In the current study, three genomic SNPs were validated as marker predictors for milk production and thermotolerance. These SNPs were in the genes of *TLR4*, *GRM8*, and *SMAD3*. Although these SNPs were located within intronic regions, they may be able to influence the expression of their corresponding genes [54]. Such ability is attributed to the presence of cis-elements or long non-coding RNAs within the intronic sequences that interact with transcription factors to regulate gene expression [55].

Toll-like receptor 4 (*TLR4*) is expressed by cells of the innate immune system in mammals, and it appears to be involved in lipid release pathways [56]. A dramatic increase in TLR4 was reported in pathogen-induced inflammatory processes related to milk production [57]. Conversely, downregulation of TLR4 cell signaling has been detected in heat-stressed dairy cows, making them more susceptible to disease [58]. Such a decrease in TLR4 induced by thermal heat was also associated with a significant reduction in lipolytic activity and lipid milk content [59]. Adipose tissue, an endocrine organ highly sensitive to climate changes, had reduced adipocyte mobilization in heat-stressed dairy cows [60]. Lactating dairy cattle under heat stress experienced a reduced responsiveness to the lipolytic effect of norepinephrine [61]. As reported by Gupta et al. [62], a close relationship has been observed between endocrine responses to heat stress and immune system activation, which appeared to involve the TLR4 signaling pathway.

In our study, results suggested that the SNP rs8193046 in the gene *TLR4* was a predictor for milk production and thermotolerance in Holstein cows because it was able to induce an immune response against heat stress, as well as modifications in lipid metabolism to minimize internal heat load. Therefore, the TLR4 gene appears to coordinate adaptive endocrine and metabolic functions in heat-stressed dairy cows, allowing them to maintain milk production.

The glutamate metabotropic receptor 8 (*GRM8*) gene encodes a presynaptic receptor that modulates glutamate release at the axon neuro-terminals. Glutamate is involved as a chemical messenger in most of the excitatory synapses and participates in the modulation of the adenylate cyclase and stimulation of phospholipase C in the cell membrane [63]. The maintenance of these functions during exposure to heat stress appears to be critical to prevent structural damaging changes and alterations in membrane fluidity [64]. Moreover, heat stress-induced glutamate activation appears to be essential for carbohydrate and amino acid metabolism; this adaptive response increases glucose synthesis, which is required to supplement the energy demands [65]. This glucose supply is the main energy source because heat stress limits lipolysis and fat mobilization to minimize the generation of heat [66].

In our study, we assumed that the SNP rs43410971 within the gene *GRM8* was validated as a genomic marker for milk production because it activated metabolic pathways allowing the heat-stressed cows to comply with their energy demands. In a similar genomic study, Cheruyoit et al. [67] reported the *GRM8* gene as a heat tolerance candidate gene that was enriched in two metabolic pathways, the neuroactive ligand–receptor interaction pathway and the glutamatergic synapse pathway. Biological pathways related to the nervous system were associated with complex adaptations in animals exposed to heat stress because these pathways allowed maintenance of a stable core body temperature by connecting the internal and external environments [68].

The SMAD family member 3 (*SMAD3*) gene encodes for an intracellular protein that is a key mediator of the TGFβ (Transforming growth factor β) signaling pathway, regulating signal transmission and transcription of their target genes [69]. In cattle, *SMAD3* mediates the inhibition of adipogenesis caused by TGFBβ through the transcriptional suppression of the PPARγ promoter [70]. In addition, *SMAD3* acts as a transcriptional regulator on the central promoter region of the *PLIN1* gene. Phosphorylation of *PLIN1* is essential for fat mobilization in adipose tissue and plays an important role in the regulation of lipid storage in adipocytes [71]. Lipogenesis in bovine primary adipocytes exposed to heat stress was reduced, which was attributed to low responsiveness to the lipogenic signals of insulin, as well as a decrease in the insulin-stimulated activation of the rate-limiting enzyme acetyl-CoA carboxylase [60].

There is strong reason to propose the SNP rs382039214 in the gene *SMAD3* as a genomic marker for milk production and thermotolerance based on its ability to regulate other genes, as well as molecular pathways highly involved in adipose tissue metabolism, specifically adipogenesis. The *SMAD3* gene appeared to be a negative regulator of adipocyte synthesis during heat stress, probably as a strategy of the body to minimize heat production. For milk production, Zhang et al. [72] constructed a miR-143 and SMAD3 regulatory network that revealed its role to increase milk fat synthesis through the formation of lipid droplets and the synthesis of triglycerides in bovine mammary epithelial cells. Such results suggested that the *SMAD3* gene reduces adipogenesis but ensures an adequate fat supply to the mammary gland in milking cows exposed to heat stress.

The allele substitution effects observed in the current study indicated that the 3 SNPs validated as genomic markers had an important contribution of the favorable allele on the traits MY305, RT, and RR. The negative estimated effects observed in the physiological variables could suggest a key role of the corresponding genes to maintain heat-stressed cows within a thermoneutral zone. These adaptive adjustments appear to favor milk production as suggested by the estimated effect of the favorable allele.

Interestingly, MY305, RT, and RR showed an improvement as the number of SNPs with favorable genotypes increased, which highlighted the positive effect of the genes *TLR4*, *GRM8,* and *SMAD3* on milk production and thermotolerance. A similar study identified one SNP from four genomic SNPs previously associated with milk yield in heat-stressed cows, as a thermotolerant marker associated with an increase in sweating rate [73]. An SNP in the *HSP90AA1* gene was reported to be associated with thermoresistance in Holstein cows, and it was proposed as a genetic marker for heat stress tolerance [74]. A recent study combining GWAS and RNAseq analysis reported the genes *PMAIP1*, *SBK1*, *TMEM33*, *GATB*, *CHORDC1*, *RTN4IP1,* and *BTBD7* as candidate markers associated with RT, RR, and drooling score in Holstein cows [21].

In our study, as the number of favorable SNP genotypes increased, Holstein cows appeared to improve their ability to maintain RT and RR at a physiological level that allowed adequate milk production. Conversely, dairy cows with no favorable genotypes appeared to be heat-sensitive, as they showed a reduction in their milk performance that was associated with an elevation in RT. Jensen et al. [75] reported that RT was 0.12 °C higher in heat-sensitive dairy cows compared to heat-tolerant cows.

Rectal temperature (RT) is a useful phenotype indicative of thermal equilibrium in animals. Cattle are able to maintain their physiological and productive functions when RT is within a thermoneutral zone. Once the RT increases above this threshold due to heat stress exposure, normal physiology is altered, leading to an increase in respiration rate (RR) and a decrease in feed intake [76]. Similarly, RR is considered a predictive marker for heat stress in dairy cattle; this physiological function allows the body to maintain temperature through evaporative cooling, acting as a powerful thermoregulatory mechanism [77].

In the current study, a negative correlation was observed between MY305 and the physiological traits RT and RR, which increased as more favorable SNP genotypes were inherited. This inverse relationship between milk yield and thermotolerance was confirmed when RT resulted as a significant predictor of MY305. Interestingly, the predictive model supporting this negative association became more powerful as the number of favorable genotypes of the genes *TLR4*, *GRM8,* and *SMAD3* increased. These results suggested that the greater the number of favorable genotypes present in the cow, the more dramatic the change in milk production will be due to variations in the RT.

Nguyen et al. [78] also reported a strong negative correlation between milk production and thermotolerance, and they suggested that selection for heat tolerance should also be included in a selection program to improve milk yield. Similarly, Jensen et al. [75] observed a reduction in milk yield and milk quality traits in dairy cows showing a high estimated breeding value for heat tolerance.

Results reported in the current study demonstrated a close association of the tested SNPs with milk yield at 305 days and thermotolerance indicators such as RT and RR. In addition, these SNPs showed a favorable contribution to improving MY305, RT, and RR. Moreover, these SNPs were in the genes *TLR4*, *GRM8*, and *SMAD3*, which appeared to be implicated in several physiological mechanisms required to maintain milk performance in dairy cows exposed to heat stress. Together, these results provided strong evidence to propose the genes *TLR4*, *GRM8*, and *SMAD* as potential molecular markers for milk production and thermotolerance in heat-stressed Holstein cows.

## 5. Conclusions

Validation of genomic markers is considered a beneficial strategy to improve our understanding of adaptive mechanisms in Holstein dairy cows, which are exposed to a heat stress environment common in semiarid or subtropical regions. In the current study, we validated three genomic SNPs in the genes *TLR4*, *GRM8*, and *SMAD3* as molecular markers for milk production and thermotolerance in heat-stressed Holstein cows. These three genes appeared to regulate metabolic processes needed to comply with energy demands and minimal heat production, which will favor the productive performance in cattle managed under extremely warm environmental conditions. Therefore, our findings may assist marker selection programs focused on the genetic improvement of milk production and thermotolerance traits in Holstein dairy cows.

## Figures and Tables

**Figure 1 biology-12-00679-f001:**
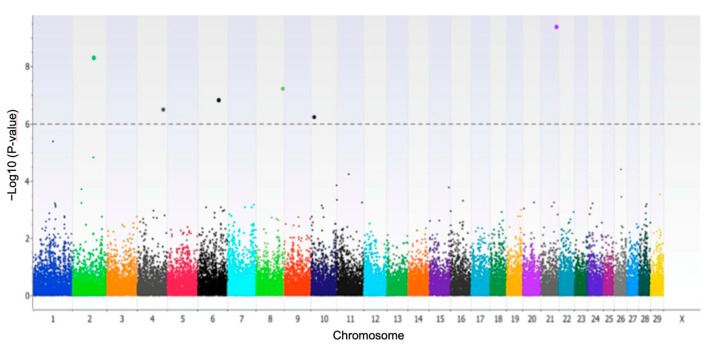
Manhattan plot displaying single-marker GWAS for total milk yield at 305 days (MY305) in dairy Holstein cows exposed to climatic heat stress.

**Figure 2 biology-12-00679-f002:**
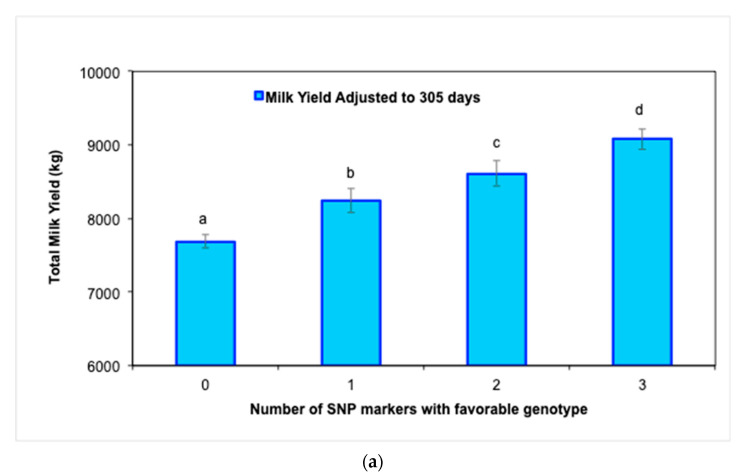
Average values (±SE) for milk production and physiological traits in Holstein cows during the experimental period according to the number of favorable genotypes (i.e., 0, 1, 2, or 3) of the three SNP markers: (**a**) milk yield adjusted to 305 days (MY305), (**b**) rectal temperature (RT), and (**c**) respiratory rate (RR) (^abcd^
*p* < 0.05).

**Figure 3 biology-12-00679-f003:**
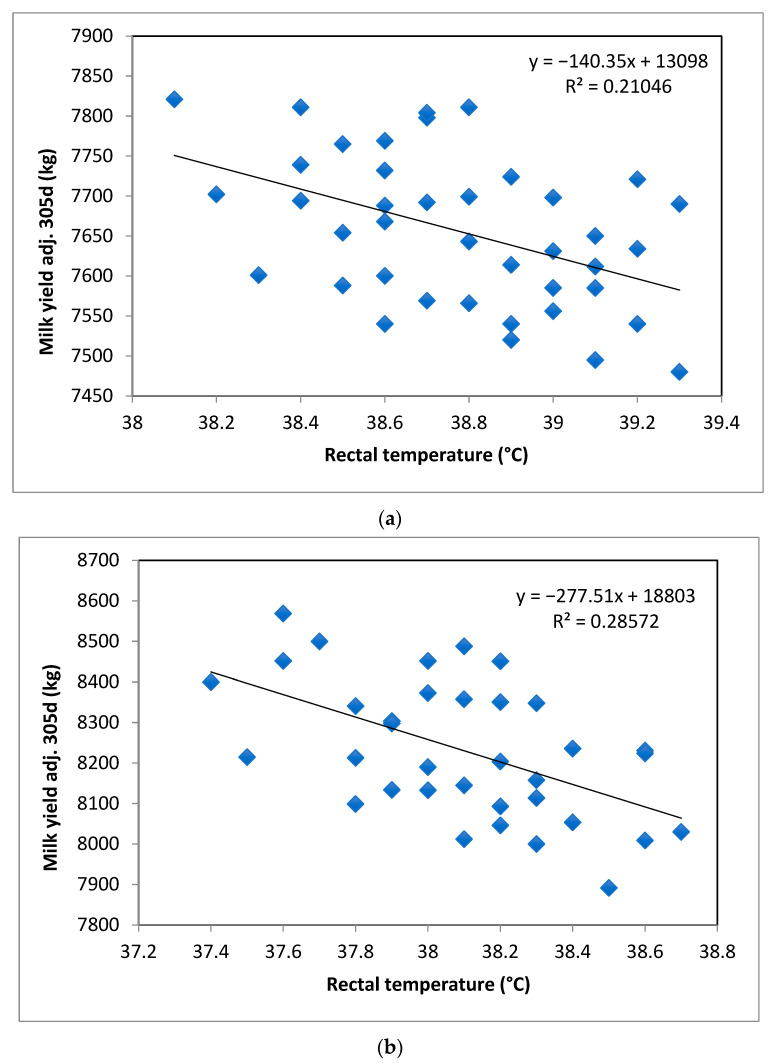
Rectal temperature (RT) as a predictor for milk yield at 305 days (MY305) according to the number of favorable genotypes of the three SNP markers: (**a**) no favorable genotypes, (**b**) one favorable genotype, and (**c**) more than one favorable genotype.

**Table 1 biology-12-00679-t001:** Summary of significant SNPs (*p* < 1.18 × 10^−6^) from single-marker genome-wide association study (GWAS) in Holstein cows managed under heat-stressed environmental conditions.

SNP ID ^1^	Variant ^2^	BTA ^3^	Position ^4^	Gene ^5^	Alleles ^6^	Variance ^7^	*p*-Value ^8^
rs109479519	Intergenic	21	60′4444453	*GLRX5*	A/G	0.240	4.33 × 10^−10^
rs29015299	Intergenic	2	82′425734	--------	T/C	0.172	5.23 × 10^−9^
rs8193046	Intronic	8	107′062912	*TLR4*	G/A	0.170	6.45 × 10^−8^
rs43410971	Intronic	4	90′731087	*GRM8*	G/A	0.159	1.21 × 10^−7^
rs108988401	Intergenic	6	80′246429	--------	A/C	0.156	1.57 × 10^−7^
rs382039214	Intronic	10	13′909494	*SMAD3*	C/T	0.150	5.96 × 10^−7^

^1^ SNP reference of the NCBI; ^2^ SNP chromosome variant; ^3^
*Bos taurus* autosomal chromosome number; ^4^ SNP position within the chromosome; ^5^ candidate gene symbol (*GLRX5* = glutaredoxin 5; *TLR4* = toll-like receptor 4; *GRM8* = glutamate metabotropic receptor; *SMAD3* = SMAD family member 3); ^6^ alleles from the SNP; ^7^ percentage of trait variance explained by the SNP; ^8^ SNP statistical significance.

**Table 2 biology-12-00679-t002:** Enriched pathways for SNP in protein-coding genes associated with milk production in cows affected by heat stress.

Canonical Pathway ^1^	*p*-Value ^2^	Key Genes ^3^
NF-kappa B signaling pathway	0.019	*CARD11/TLR4/NFKB1*
PI3K-Akt signaling pathway	0.022	*TCL1A/TLR4/NFKB1*
TGF-beta signaling pathway	0.035	*SMAD3/SMAD6*
Toll-like receptor signaling pathway	0.041	*TLR4/NFKB1*
Glutamatergic synapse	0.042	*GRM8/TRPC1*

^1^ Significant pathways for candidate genes associated with MY305; ^2^ *p*-value adjusted according to the Benjamini–Hochberg correction; ^3^ Candidate gene names.

**Table 3 biology-12-00679-t003:** Identification, gene name, favorable SNP allele, allele frequencies, and Hardy–Weinberg equilibrium analysis for genomic SNPs associated with milk and thermotolerance phenotypes.

SNP ID ^1^	Gene ^2^	F. Allele ^3^	Allele Frequency ^4^		HWE Test ^5^	HWE *p*-Value ^6^
			A	G		
rs109479519	*GLRX5*	A	0.43	0.57	26.31	<0.0001
rs8193046	*TLR4*	A	0.55	0.45	0.14	0.61
rs43410971	*GRM8*	G	0.28	0.72	2.16	0.23
			C	T		
rs382039214	*SMAD3*	T	0.16	0.84	0.94	0.38

^1^ SNP reference of the NCBI; ^2^ gene symbol name (*TLR4* = toll-like receptor 4, *GRM8* = glutamate metabotropic receptor 8, *SMAD3* = SMAD family member 3); ^3^ allele with the favorable effect on phenotype; ^4^ frequency of both alleles within cow population; ^5^ Hardy–Weinberg equilibrium “χ^2^” test value; ^6^ “χ^2^” test *p*-value with 1 degree of freedom and α = 0.05.

**Table 4 biology-12-00679-t004:** Least-square means ± SE according to SNP’s marker genotypes for milk yield and physiological traits in Holstein cattle validation populations.

SNP ID ^1^	Trait ^2^	Least-Square Means by Genotype ± SE ^3^			*p*-Value ^4^
		AA	AG	GG	
rs8193046	MY305	8794.88 ± 35.33 ^a^	8453.02 ± 39.75 ^b^	7826.04 ± 39.29 ^c^	<0.0001
	RT	37.69 ± 0.05 ^a^	38.01 ± 0.04 ^b^	38.62 ± 0.05 ^c^	<0.0001
	RR	61.02 ± 0.41 ^a^	69.02 ± 0.49 ^b^	75.25 ± 0.38 ^c^	<0.0001
rs43410971	MY305	7645.21 ± 38.76 ^a^	8120.55 ± 37.21 ^b^	8861.79 ± 38.18 ^c^	<0.0001
	RT	38.47 ± 0.06 ^a^	37.96 ± 0.05 ^b^	37.55 ± 0.04 ^c^	<0.0001
	RR	74.32 ± 0.47 ^a^	66.93 ± 0.46 ^b^	60.52 ± 0.42 ^c^	<0.0001
		CC	CT	TT	
rs382039214	MY305	7743.15 ± 38.76 ^a^	8501.30 ± 37.21 ^b^	8712.44 ± 34.18 ^b^	0.0009
	RT	38.26 ± 0.06 ^a^	37.42 ± 0.05 ^b^	37.31 ± 0.04 ^b^	0.0037
	RR	70.14 ± 0.39 ^a^	67.17 ± 0.42 ^a^	63.29 ± 0.47 ^a^	0.0952

^1^ SNP reference of the NCBI; ^2^ phenotypic traits (MY395 = total milk yield adjusted to 305 days, kg; RT = rectal temperature, °C; RR = respiration rate, breaths/min); ^3^ least-square means (*TLR4* = toll-like receptor 4, *GRM8* = glutamate metabotropic receptor 8, *SMAD3* = SMAD family member 3); least-square means according to SNP genotype ± SE (^a,b,c^ indicate statistical difference among least-square means by genotype in the mixed model at *p* < 0.05); ^4^ *p*-value = statistical significance.

**Table 5 biology-12-00679-t005:** Allele substitution effects and fixed estimates for additive and dominance effects of the favorable allele for MY305, RT, and RR in the validation Holstein cattle populations.

SNP ID ^1^	Trait ^2^	Allele Substitution Effects			Fixed Estimates Effects		
		F. Allele ^3^	*p*-Value ^4^	Estimate ± SE ^5^	*p*-Value ^6^	AddE ^7^	DomE ^8^
rs8193046	MY305	A	<0.01	570.57 ± 26.94	<0.01	574.42	52.56
	RT	A	<0.01	−0.43 ± 0.04	<0.01	0.46	0.15
	RR	A	<0.01	−7.06 ± 0.29	<0.01	7.12	0.88
rs43410971	MY305	G	<0.01	602.18 ± 38.17	<0.01	608.29	132.95
	RT	G	<0.01	−0.41 ± 0.05	<0.01	0.46	0.05
	RR	G	<0.01	−6.72 ± 0.44	<0.01	6.91	0.49
rs382039214	MY305	T	<0.01	476.22 ± 35.14	<0.01	484.65	273.51
	RT	T	<0.01	−0.47 ± 0.05	<0.01	0.47	0.36
	RR	T	0.12	−5.69 ± 0.43	0.09	3.43	0.45

^1^ SNP reference of the NCBI; ^2^ phenotypic traits (MY395 = total milk yield adjusted to 305 days, kg; RT = rectal temperature, °C; RR = respiration rate, breaths/min); ^3^ SNP allele with a favorable effect on the phenotype; ^4^ *p*-values obtained from allele substitution analysis in SAS which included the term genotype as a covariate; ^5^ estimates of the effect expressed in units of the traits ± standard error; ^6^ *p*-values for fixed effects were obtained from the substitution of favorable allele analysis that included the genotype term as fixed effect; ^7^ the additive effect was estimated as the difference between the 2 homozygous means divided by 2; ^8^ the dominance effect was calculated as the deviation of the heterozygous from the mean of the 2 homozygous.

**Table 6 biology-12-00679-t006:** Pearson correlations between physiological variables and milk yield at 305 days (MY305) according to the number of favorable genotypes from the 3 significant SNP markers.

Physiological Trait	Number of Favorable SNP Genotype Markers		
	0	1	2–3
Rectal temperature (RT)	−0.4587 **	−0.5345 **	−0.6021 **
Respiratory rate (RR)	−0.2905 *	−0.3139 *	−0.3868 *

** values are highly significant at *p* < 0.01; * values are significant at *p* < 0.05.

## Data Availability

The data that support the findings of this study are available from the corresponding author, P.L.-N., upon reasonable request.

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
