# Peer review of "Genetic Markers Associated with Milk Production and Thermotolerance in Holstein Dairy Cows Managed in a Heat-Stressed Environment"

_biology, 2023, doi:10.3390/biology12050679_

Round 1
Reviewer 1 Report
Zamorano-Algandar et al. reported some genetic markers for milk production and thermotolerance in Holstein dairy cows managed in a heat-stressed environment. Overall, this study is well designed, and methods are reasonable. The authors also performed the validation for GWAS results which makes the manuscript more complete. One of the limitations is the small sample size for GWAs as n =300.
Line 32: I think it is medium density chip, high density is 700K.
Line 34: Change 10-6 to 10-6
n=216 for both populations or n=216 for each independent population
Line 78-79: Might give some information about the number of QTLs for the traits and add the accessed date for QTL database.
Line 145: Did the authors check the effects of DGAT1 mutation (K232A (Lys232 → Ala)) on MY305?
Line 145: It is not clear what is g in the models? What are the assumptions for the effects?
How many sires from these animals?
Table 1: Why did the authors report the values greater than significant threshold of p
Table 2: How many genes did the authors use for enrichment analyses, the pathways enriched with only one gene in the gene list does not make sense.
Table 3 and in the manuscript: gene symbols should be in Italics.
Table 4 and 5: Might add the unit for the traits.
The discussion is fine, I suggest the authors adding some discussion of limitation of GWAS (low sample size) and about the DGAT1 mutation,
Author Response
Dear reviewer:
Thank you for your valuable comments and suggestions that helped to improve the quality of the manuscript. We have answered to each individual comments and suggestions, and responses point by point are described below:
Zamorano-Algandar et al. reported some genetic markers for milk production and thermotolerance in Holstein dairy cows managed in a heat-stressed environment. Overall, this study is well designed, and methods are reasonable. The authors also performed the validation for GWAS results, which makes the manuscript more complete. One of the limitations is the small sample size for GWAS as n=300.
- Line 32: I think it is medium density chip, high density is 700 k.
Response: We have made this correction.
- Line 34: Change 10-6 to 10-6
Response: We have made this correction.
- n=216 for both populations or n=216 for each independent population?
Response: n=216 for both populations. We have made this clarification in the Abstract (page 1) and Materials and Methods section (page 4).
- Line 78-79: Might give some information about the number of QTLs for the traits and add the accessed date for QTL database
Response= We summarized in the Introduction section (page 2) the QTLs reported by Trait Classes in the “Cattle QTLdb data summary”, and indicated the accessed date.
- Line 145: Did the authors check the effects of DGAT1 mutation (K232A (Lys232 → Ala)) on MY305?
Response: We did not find this mutation as marker in our study; it maybe due to the severe heat stress at which Holstein cows were exposed during the study, or due to the small sample size. We included some references in Discussion section (page 13) that reported the association of the DGAT1 mutation K232A with milk production traits in Holstein cattle.
- Line 145: It is not clear what is g in the models? What are the assumptions for the effects?
Response: We have clarified the genomic model in Materials and Methods section and we described the assumptions for the effects (page 4).
- How many sires from these animals?
Response: We have added this information in Materials and Methods section (page 4).
- Table 1: Why did the authors report the values greater than significant threshold of p
Response: We have made a correction in Table 1, which now only report SNPs with values lower than significant threshold of p.
- Table 2: How many genes did the authors use for enrichment analyses, the pathways enriched with only one gene in the gene list does not make sense.
Response: We used for enrichment analyses 36 candidate genes, which resulted significant at P less than 0.01. We added this information in Materials and Methods section (page 4). We also updated Table 2 by including all enriched genes in each pathway; because the previous version of this table only included enriched genes with a P-value lower than the significant threshold (P < 1.18 x 10-6).
- Table 3 and in the manuscript: gene symbols should be in Italics.
Response: We have made this correction in Table 3 and throughout the manuscript.
- Table 4 and 5: Might add the unit for the traits.
Response: We have made this correction in Tables 4 and 5.
- The discussion is fine, I suggest the authors adding some discussion of limitation of GWAS (low sample size) and about the DGAT1 mutation.
Response: We have added in Discussion section some information about limitations of GWAS due to low sample size (page 13), as well as several references that included information related to DGAT1 mutation in dairy Holstein cows (page 13).
Reviewer 2 Report
The present research is to "Validate molecular markers associated with milk production and thermotolerance traits in Holstein cows managed in a warm semi-arid region." This research provides interesting information. However, it is necessary to make some important changes before its final publication.
INTRODUCTION
Expand this section further with respect to “SNP”, “genetic marker effect in milk”
MATERIAL AND METHODS
General comments: I recommend mentioning the type of production system and what were the selection criteria for these animals. In addition, a factor of increased temperature in the animal is related to rumen fermentation that increases the temperature of the cow due to a lower efficiency in the dissipation of this heat, why did they not evaluate feed consumption as a variable?
Line 103.- mentions “Three hundred Holstein cows calving in spring, from 4 to 6 years old. older and in good body condition (3.0 - 4.0) were included in this study”. It would be convenient to mention the average weight and body condition for each group, as well as "± SE".
Line 120-123.- It mentions "The ambient temperature (AT; °C) and relative humidity (RH; %) data were collected by a nearby weather station (Red de Estaciones Meteorológicas Automáticas de Sonora) and extracted from the website of REMAS (http://www.siafeson.com/remas). Daily records of AT and HR were used to calculate the temperature-humidity index (THI) using the formula THI = (0.8 × AT) + [(HR/100) × (AT − 14.4)] + 46, 4 [23]”. In which section of the results is the data reported or related to the “THI”?
Line 180.- It is mentioned “The cows included in the validation study calved in spring, from 4 to 6 years of age. age and body condition between 2.5 and 4.0”. This is already mentioned in line 103 “Three hundred Holstein cows calving in spring, 4 to 6 years old. older and in good body condition (3.0 - 4.0) were included in this study”. Recommend removing it.
Line 183-186.- mentions “Physiological traits of rectal temperature (RT) and respiratory rate (RR) were measured bi-weekly (06:00 and 16:00 hours) as indicators of thermotolerance. A digital thermometer was used to record RT, whereas RR was evaluated by counting the flank movements for 15 s (breaths/min).” I recommend mentioning how they made these measurements? Who made these measurements? Why did they only consider measuring (06:00 and 16:00 hours)? And finally, because they measured these physiological parameters for only 15 seconds, I mention this because I consider it necessary to have measured it for a minute, not in a small amount of time. This is because if there is a decrease or increase during the first 15 seconds, it will affect the final result and therefore, the interpretation of these data.
Line 213.- It would be convenient to mention if the data came from a population with normal distribution or if they were transformed. How were the data represented?
Line 236.- I was mentioned “One-way ANOVA was used to compare MY305, RT and RR in cows according to the number of favorable SNP genotypes”. Mention if the data used for this test belonged to a Normal and homoscedastic population?
DISCUSSION
In general, I recommend guiding the discussion according to how the results were reported. I also recommend paying more attention to describing the possible implications of the genes “TLR4, GRM8 and SMAD3 as molecular markers for milk production and thermotolerance”.
Line 441.- mentions “This region is characterized by extended periods of high ambient temperature and high relative humidity throughout the year, which elevates the THI value above the threshold of units in middle-spring.” I recommend avoiding comparing with "THI" values because remember that this result represents the index that relates environmental temperature with respect to humidity and not the temperature of the animal. They can carry out the conversion if they have the data as mentioned in the "Materials and methods" section, but this would have to be reported first in the results.
Author Response
Dear reviewer:
Thank you for your valuable comments and suggestions that helped to improve the quality of the manuscript. We have answered to each individual comments and suggestions, and responses point by point are described below:
The present research is to "Validate molecular markers associated with milk production and thermotolerance traits in Holstein cows managed in a warm semi-arid region." This research provides interesting information. However, it is necessary to make some important changes before its final publication.
INTRODUCTION
-Expand this section further with respect to “SNP”, “genetic marker effect in milk”
Response: We added information in the Introduction section (page 2) with respect to “SNP” definition, and also added an interesting example of a SNP/gene with a “genetic marker effect in milk”.
MATERIAL AND METHODS
- General comments: I recommend mentioning the type of production system and what were the selection criteria for these animals. In addition, a factor of increased temperature in the animal is related to rumen fermentation that increases the temperature of the cow due to a lower efficiency in the dissipation of this heat, why did they not evaluate feed consumption as a variable?
Response: We have added in Materials and Methods section (page 3) information about the type of production system, as well as selection criteria for cows included in the study.
Regarding to feed consumption, we collected data about daily feed consumed per pen, but we do not have individual feeders to calculate feed consumed per cow. Therefore, we do not have available for genomic evaluation the variable “feed consumption”. I apologize by that.
- Line 103.- mentions “Three hundred Holstein cows calving in spring, from 4 to 6 years old and in good body condition (3.0 - 4.0) were included in this study”. It would be convenient to mention the average weight and body condition for each group, as well as "± SE".
Response: We have added information about average weight and body condition, as well as their standard error in Materials and Methods section (page 3).
- Line 120-123.- It mentions "The ambient temperature (AT; °C) and relative humidity (RH; %) data were collected by a nearby weather station (Red de Estaciones Meteorológicas Automáticas de Sonora) and extracted from the website of REMAS (http://www.siafeson.com/remas). Daily records of AT and HR were used to calculate the temperature-humidity index (THI) using the formula THI = (0.8 × AT) + [(HR/100) × (AT − 14.4)] + 46, 4 [23]”. In which section of the results is the data reported or related to the “THI”?
Response: We have added a new subheading in Results section (3.1. Climatic conditions) in which the data related to the “THI” were included (page 6).
- Line 180.- It is mentioned “The cows included in the validation study calved in spring, from 4 to 6 years of age. age and body condition between 2.5 and 4.0”. This is already mentioned in line 103 “Three hundred Holstein cows calving in spring, 4 to 6 years old and in good body condition (3.0 - 4.0) were included in this study”. Recommend removing it.
Response: I kindly inform the reviewer that this data refers to the validation population (n=216), which is different to the initial population used for GWAS that was described in line 103 (n=300). I would suggest do not remove this information to make clear that we used two different Holstein cow populations during the study.
- Line 183-186.- mentions “Physiological traits of rectal temperature (RT) and respiratory rate (RR) were measured bi-weekly (06:00 and 16:00 hours) as indicators of thermotolerance. A digital thermometer was used to record RT, whereas RR was evaluated by counting the flank movements for 15 s (breaths/min).” I recommend mentioning how they made these measurements? Who made these measurements? Why did they only consider measuring (06:00 and 16:00 hours)?
Response: We have added in Materials and Methods section (page 5) the information related to how RT and RR were evaluated, who did these measurements and why RT and RR were only collected at 06:00 and 16:00 hour.
- And finally, because they measured these physiological parameters for only 15 seconds, I mention this because I consider it necessary to have measured it for a minute, not in a small amount of time. This is because if there is a decrease or increase during the first 15 seconds, it will affect the final result and therefore, the interpretation of these data.
Response: I apologize because we actually collected respiration rate data trough measuring the number of breaths during 60 seconds at 06:00 and 16:00 hours, and these is the data that we reported in the manuscript. However, due to an unintentional mistake on my part I mentioned that RR data were collected during 15 seconds. I have added this correction in Materials and Methods section (page 5).
- Line 213.- It would be convenient to mention if the data came from a population with normal distribution or if they were transformed. How were the data represented?
Response: We have clarified in Materials and Methods section that the data come from a population with normal distribution, and we have added information about data representation (page 5).
- Line 236.- I was mentioned “One-way ANOVA was used to compare MY305, RT and RR in cows according to the number of favorable SNP genotypes”. Mention if the data used for this test belonged to a Normal and homoscedastic population?
Response: We have now mentioned in Materials and Methods section (page 6) that the data used for the ANOVA analysis belonged to a Normal and homoscedastic population.
DISCUSSION
- In general, I recommend guiding the discussion according to how the results were reported. I also recommend paying more attention to describing the possible implications of the genes “TLR4, GRM8 and SMAD3 as molecular markers for milk production and thermotolerance”.
Response: We have arranged some paragraphs of the Discussion (page 12) to follow an order, according to how the results were reported: climatic data, genomic analysis, SNP validation study, and confirmation of SNPs thermotolerance ability.
In Discussion section we described the possible physiological function of the genes TLR4, GRM8 and SMAD3, and explained why we consider these genes as molecular markers for milk production and thermotolerance. In addition, we have included a paragraph at the end of the Discussion section about general implications of theses genes as molecular markers.
- Line 441.- mentions “This region is characterized by extended periods of high ambient temperature and high relative humidity throughout the year, which elevates the THI value above the threshold of units in middle-spring.” I recommend avoiding comparing with "THI" values because remember that this result represents the index that relates environmental temperature with respect to humidity and not the temperature of the animal. They can carry out the conversion if they have the data as mentioned in the "Materials and methods" section, but this would have to be reported first in the results.
Response: We have corrected this paragraph avoiding “THI” comparisons. Instead we only mentioned that the high THI during warm seasons is able to create heat stress conditions for Holstein cattle. Moreover, we have added at the beginning of the Results section the data about THI values during the experimental period.